# RSBagging: An ensemble classifier detecting the after-effects of ischemic stroke through EEG connectivity and microstates

**Fang Wang**[1]*, **Xueying Zhang**[2], **Peng Zhang**[3], **Fengyun Hu**[3]*

1 School of Big Data and Artificial Intelligence, Chengdu Technological University, Chengdu, China, 2 College of Information and Computer, Taiyuan University of Technology, Taiyuan, China, 3 Department of Neurology, Shanxi Provincial People's Hospital affiliated with Shanxi Medical University, Taiyuan, China

* wang_fang_ty@163.com (FW); fengyun71@163.com (FH)

## Abstract

### Background and purpose

Stroke can lead to significant after-effects, including motor function impairments, language impairments (aphasia), disorders of consciousness (DoC), and cognitive deficits. Computer-aided analysis of EEG connectivity matrices and microstates from bedside EEG monitoring can replace traditional clinical observation methods, offering an automatic approach to monitoring the progression of these after-effects. This EEG-based method also enables quicker and more efficient assessments for medical practitioners.

### Methods

In this study, we employed Functional Connectivity features that extract spatial representation and Microstate features that focus on the time domain representation to monitor the after-effects of ischemic stroke patients. As the dataset from stroke patients is heavily imbalanced across various clinical after-effects conditions, we designed an ensemble classifier, RSBagging, to address the issue of classifiers often favoring the majority classes in the classification of imbalanced datasets.

### Results

The experimental results demonstrate that different connectivity matrices are effective for three classification tasks: consciousness level, motor disturbance, and stroke location. Using our RSBagging model, all three tasks achieve over 98% accuracy, sensitivity, specificity, and F1-score, significantly outperforming the existing classifiers SVM, XGBoost, and Random Forest.

### Conclusion

Therefore, the RSBagging classifier based on connectivity matrices offers an effective method for monitoring the after-effects in stroke patients.

**Data Availability Statement:** The data, along with the extracted features and labels, can be found at https://github.com/linda-edward/RSBagging.

**Funding:** The author(s) received no specific funding for this work.

**Competing interests:** The authors have declared that no competing interests exist.

## Introduction

Stroke is a prevalent neurological disorder worldwide, constituting a leading cause of severe disability. It often results in significant after-effects, such as extensive motor function impairment [1, 2], disorder of consciousness (DoC) [3], aphasia and cognitive dysfunction [4]. These disabilities seriously affect patients' quality of life. Resting-state electroencephalography (EEG) monitoring is an effective way to assist medical practitioners in rapidly assessing after-effects of stroke. Studies have found that EEG indices of sub-acute ischemic stroke are correlated with traditional clinical scores, such as the NIHSS, and may inform future management of stroke patients [5]. The investigation of the correlation between early EEG biomarkers and functional and morphological outcomes in thrombolysis-treated strokes helps better establish the treatment strategies [6, 7]. A recent EEG fractal analysis indicates that stroke patients have significantly less complex brain activity compared to healthy individuals during the acute and early subacute stages [8].

Brain functional connectivity and EEG microstates have demonstrated associations with specific phenotypes. Brain functional connectivity, which refers to the correlation patterns between different brain regions, has been linked to various psychiatric, behavioral, and cognitive characteristics [9, 10]. Several connectivity matrices have been developed, focusing on the amplitude and phase of oscillatory activity across different frequency bands, and have been correlated with diverse phenotypes. For instance, the phase-locking value (PLV) has been instrumental in emotion recognition [11] and in differentiating schizophrenia [12]; the weighted phase lag index (wPLI) distinguishes wakefulness from sleep [13]; whereas the Pearson correlation coefficient (PCC) and coherence (COH) detect Disorders of Consciousness (DoC) in brain injuries [14]. EEG microstates, which are brief semi-stable periods in global electrical brain activity captured by EEG scalp recordings, have also been associated with specific phenotypes. Microstate analysis segments EEG data into a limited number of clusters, each lasting 40-150 ms, based on global scalp points [15]. These transient periods of stability manifest as distinct topographical representations known as microstates. Numerous studies explore how EEG microstate properties vary across cognitive tasks, genders, medications, and diseases. For example, a study in NeuroImage examined how the spatial and temporal properties of microstates might be altered by manipulating cognitive tasks (a serial subtraction task versus wakeful rest) [16]. Changes in EEG microstates have also been explored in patients with head injury [17], narcolepsy [18], Alzheimer's disease [19] and stroke [1].

Recent years have seen a surge in published articles utilizing machine learning techniques, leveraging various features and biomarkers extracted from signal processing and brain connectivity analysis. Two prevalent methods for constructing these models are: (a) Ensembles of decision trees (e.g., random forests and gradient boosting machines) for structured data, and (b) Multilayered neural networks trained with Stochastic Gradient Descent (i.e., shallow and/ or deep learning) for unstructured data [20]. Given the structured nature of the EEG data in this study, involving connectivity matrices and EEG microstates, an ensemble classifier is deemed more suitable than multilayered neural networks. Additionally, ensemble classifiers offer advantages such as interpretability, faster training, independence from specialized GPU hardware for large-scale inference, and reduced hyperparameter tuning compared to deep learning methods. Notably, prominent ensemble classifiers include Adaboost [21], Xgboost [22], and Random Forest, all constructed by aggregating decision trees through bagging or boosting techniques.

Despite the superior accuracy of these boosting classifiers on balanced datasets, our experiments reveal their limitations in effectively detecting typical phenotypes within real clinical datasets, often characterized by imbalanced class distributions. For instance, in this study,

disorders of consciousness and motor disturbances are challenging to detect due to their imbalanced representation. Even with resampling techniques like random under-sampling or over-sampling, the enhancements achieved by existing ensemble classifiers remain modest. Addressing imbalanced datasets, skewed toward one or more classes, poses a significant challenge for classifiers.

To mitigate the drawbacks of existing boosting methods, this paper introduces a novel ensemble classifier called Bagging of Support Vector Machines with Random Under-Sampling (RSBagging). Our RSBagging model utilizes resampling techniques to tackle classifier bias towards the majority class and injects diversity into each base classifier through random under-sampling with replacement. Initially, the original imbalanced dataset is divided into training and testing sets. Subsequently, the training set is further subdivided into multiple balanced training subsets using random under-sampling. Each independent base classifier SVM is trained on an imbalanced training subset and provides predictions for the same test set. Finally, majority voting is employed to aggregate the predictions from the multiple base SVM models.

The remainder of this paper is structured as follows: we begin with the materials and feature extraction section, followed by an exposition on the RSBagging ensemble classifier model. Subsequently, we present the classification results and discussions pertaining to three classification tasks. Finally, the Conclusion section provides a summary and closing remarks.

## Materials and feature extraction

### Participants and clinical assessment

Table 1 summarized the demographics and clinical characteristics of the participants. There were 241 stroke patients (mean age = 66.23 years, standard deviation (SD) = 13.17 years) in this study. These subjects were patients admitted to the neurology department at Shanxi Provincial People's Hospital after acute stroke. Inclusion criteria were as follows: (1) The patients were diagnosed with ischemic stroke, (2) EEG data were recorded and available for analysis, and (3) the corresponding assessment of consciousness by medical practitioners was recorded. Exclusion criteria were: (1) patients younger than 18 years old, and (2) pregnant patients.

In our study, the state of consciousness, stroke location (from MRI), and motor disturbances (from clinical assessments) were recorded for all participants, as summarized in Table 1. Neurological examinations assessing consciousness and motor disturbances were conducted immediately before EEG signal recordings. Stroke locations were identified via MRI, and motor disturbances were diagnosed by clinicians. All clinical assessments were performed by medical practitioners blinded to the EEG measures.

**Table 1. Demographics and clinical characteristics of 241 patients.**

| State of awake or consciousness | Number of subjects (Female/Male) | Age (Mean±SD) | Stroke location (MRI) #subjects (B / L / R) | Motor disturbance #subjects (B / L / R) |
|---|---|---|---|---|
| Awake Consciousness | 37 / 107 | 64.52±12.80 | 36 / 52 / 47 | 23 / 37 / 46 |
| Somnolence | 18 / 34 | 71.31±12.09 | 9 / 29 / 11 | 15 / 10 / 20 |
| Stupor | 13 / 12 | 67.88±13.63 | 8 / 10 / 6 | 9 / 5 / 9 |
| Light coma | 3 / 12 | 64.60±16.28 | 6 / 4 / 4 | 8 / 4 / 2 |
| Middle coma | 0 / 2 | 69.00±0.00 | 1 / 1 / 0 | 1 / 0 / 1 |
| Deep coma | 1 / 2 | 53.33±9.24 | 0 / 0 / 0 | 0 / 0 / 1 |
| OVERALL | 72 / 169 | 66.23±13.17 | 60 / 96 / 68 | 56 / 56 / 79 |

All data were collected as part of a retrospective study of medical records, approved by the local institutional review board of Shanxi Provincial People's Hospital and the ethics committee waived the need for consent. The data were accessed for research purposes from January 16, 2024, to January 15, 2025. The authors had no access to information that could identify individual participants during or after data collection. All methods in this study were carried out in accordance with relevant guidelines and regulations.

## EEG data acquisition and pre-processing

EEG data were gathered using a bedside digital video EEG monitoring system (Solar 2000 N, Solar Electronic Technologies Co., Ltd, Beijing, China) at a sampling rate of 500 Hz. In line with the International Federation of Societies for Electroencephalography and Clinical Neurophysiology recommendations, we adhered to the international 10-20 system for electrode placement. This involved positioning 20 Ag/AgCl electrodes, as depicted in Fig 1.

EEG preprocessing was performed using MATLAB (MathWorks, Natick, MA) and the EEGLAB (version 14.1.1b) and Fieldtrip toolboxes. For analysis, we selected ten minutes of EEG data from each participant. The preprocessing steps included:

(a) Identifying and interpolating faulty EEG channels based on neighboring channel statistics.

(b) Re-referencing all channel signals to an average reference.

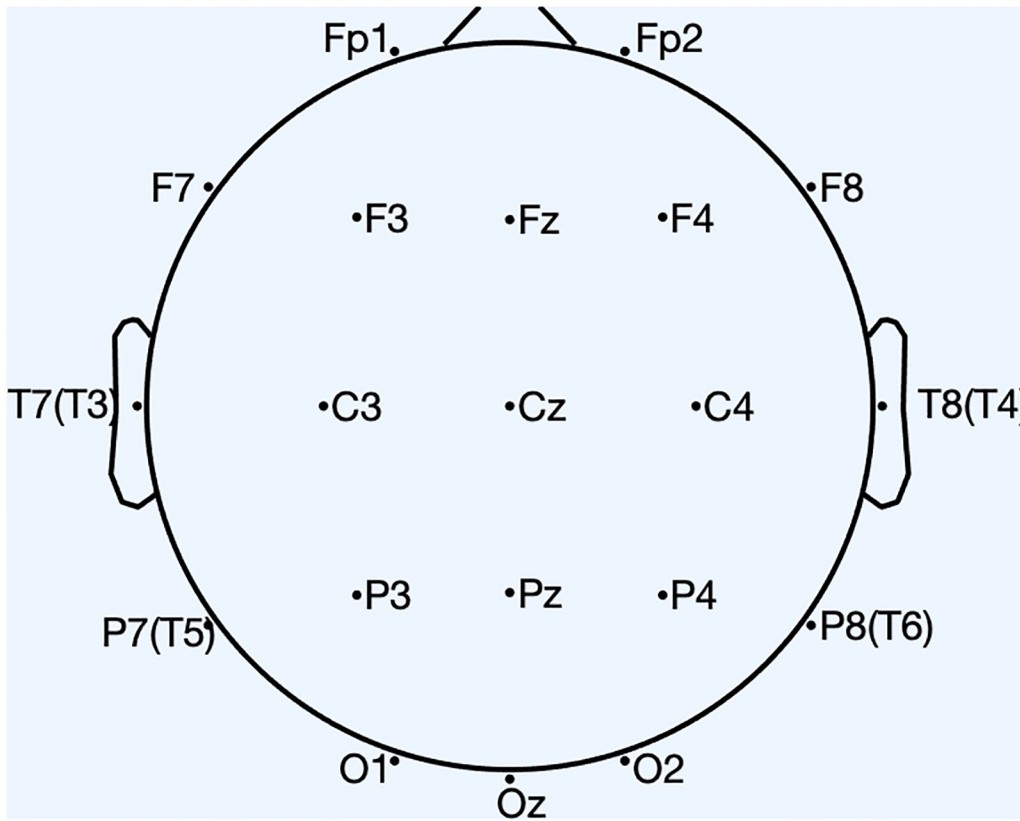

**Fig 1. EEG channel locations.**

(c) Applying high-pass filtering at 0.5 Hz and low-pass filtering at 45 Hz using a finite impulse response (FIR) filter.

(d) Using the automatic continuous rejection tool in EEGLAB to detect and eliminate artifacts.

### Brain connectivity feature extraction

Brain connectivity can be categorized into three types: structural, functional, and effective connectivity [23]. Structural connectivity refers to the physical connections between neurons, such as the axonal connections between one neural mass and another which is outside the scope of this study. Here, we focus on functional and effective connectivity. Functional connectivity measures the correlation between different regions of the brain, while effective connectivity measures the causal relationships between regions, i.e. how changes in one region lead to changes in another [24].

The coordination of information flow between brain regions is essential for performing various cognitive and perceptual tasks. The brain can adjust the flow of information by altering the strength, pattern, or frequency of oscillatory synchrony between different brain areas [25]. To describe the relationship between brain regions, researchers use various connectivity matrices, however, each connectivity matrices has advantages and disadvantages, and its vigorous adherents and opponents [23]. As a result, it is often difficult to know which connectivity matrice is effective for specific tasks.

This study considers 11 connectivity matrices from stroke patients' EEG signals to explore the most effective ones for the task of detecting DoC, motor disturbance and stroke location side of the brain in stroke patients, respectively. The connectivity matrices are calculated in MATLAB using the FieldTrip Toolbox. The measures conclude connectivity matrices of synchronization and multivariate spectral decomposition. These matrices are computed from frequency from 1Hz to 45Hz and then averaged before input into classifiers. Each connectivity matrix has a shape of 190. Fig 2 shows the flowchart of calculation of the 11 connectivity matrices extracted in this study.

**Nonparametric methods.** Nonparametric methods includes linear methods and nonlinear methods.

One widely used linear metric is the COH [26]. It is a normalized linear measure of correlations between two time series signals within a certain frequency band which is sensitive to changes in power and also changes in phase relationships [27]. The COH is defined as:

$$COH = \frac{|S_{x_1 x_2}(f)|^2}{S_{x_1 x_1}(f) S_{x_2 x_2}(f)} \tag{1}$$

where $S_{x_1 x_2}(f)$ is the cross power spectral density of two EEG channels $x_1(t)$ and $x_2(t)$, using Welch's averaged, modified periodogram method of spectral estimation; $S_{x_1 x_1}(f)$ and $S_{x_2 x_2}(f)$ are the individual power spectral densities of $x_1(t)$ and $x_2(t)$, respectively.

The following metrics are nonlinear methods.

The PLV estimates how the relative phase is distributed over the unit circle [12, 27]. PLV, taking the absolute average of phase differences over temporal windows, can be computed as below:

$$PLV = \frac{1}{N} \left| \sum_{k=0}^{N-1} e^{i\Delta\phi(t_k)} \right| \tag{2}$$

where $\Delta\phi$ represents the phase difference of a pair of electrodes, $x_1(t)$ and $x_2(t)$ (here, $\Delta\phi$ is

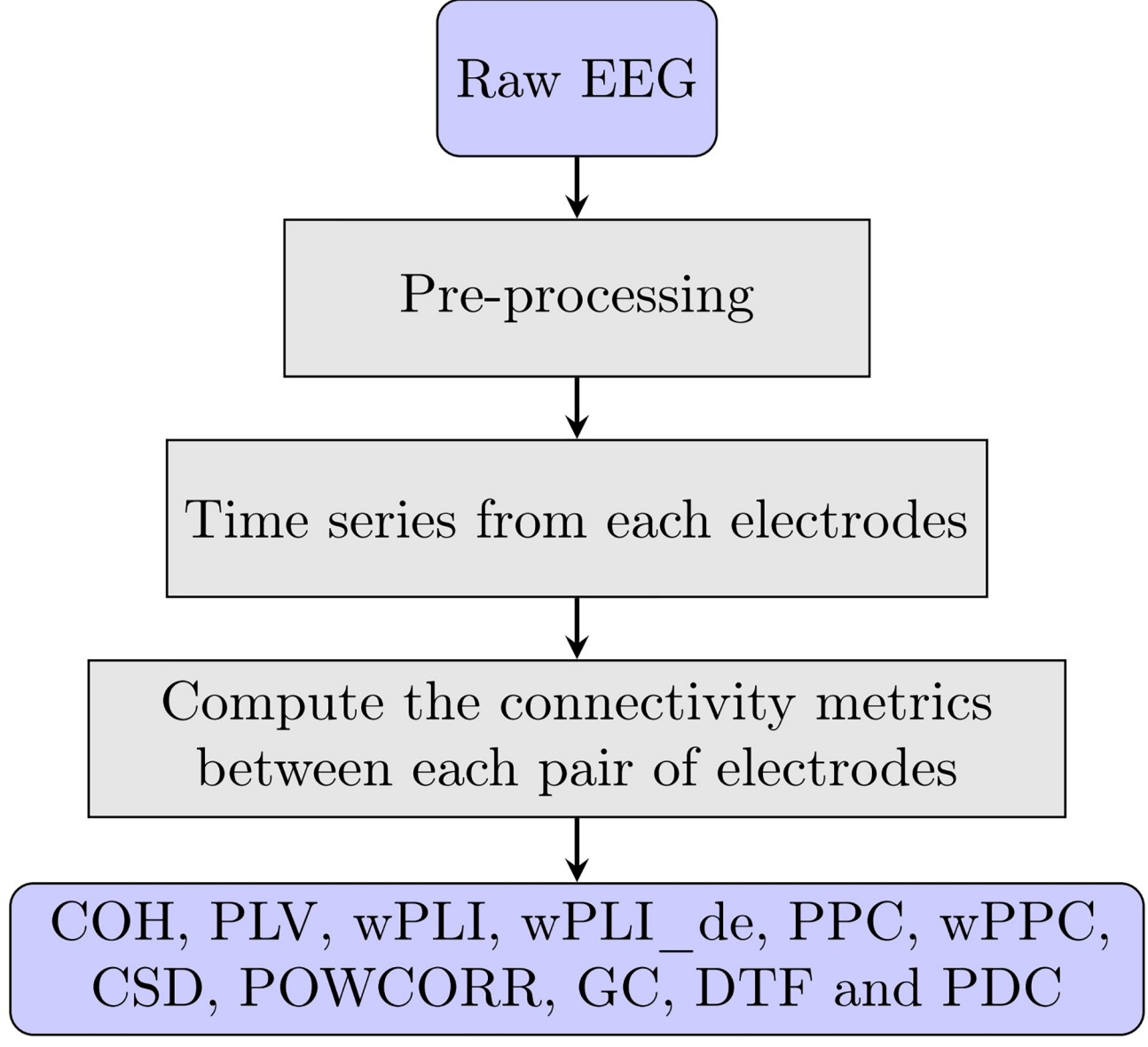

**Fig 2. The calculation flowchart of connectivity matrices.**

calculated in the same way as our previous study [28]); $i$ represents the imaginary unit; $t_k$ represents the $k$th discrete time-step.

The weighted PLI (wPLI) and debiased weighted PLI (wPLI_de) are adjustments to the PLI. The phase lag index (PLI) is a metric that values the distribution of phase differences across observations [29]. It is computed by averaging the sign of the per observation estimated phase difference [25]. The wPLI and wPLI_de are more robust against field spread, noise and sample-size bias than PLI.

The pairwise phase consistency (PPC) and weighted pairwise phase consistency (wPPC) are computed from a data matrix containing a cross-spectral density. PPC is equivalent to the population statistic of the squared PLV [30]. The weighting of the wPPC is according to the magnitude of the cross-spectrum.

The above five metrics provide information about phase synchronization, focusing on the phase coupling of oscillatory systems.

The cross-spectral density (CSD) analyzes the relationship between two signals in the frequency domain [31]. It determines whether the peaks at the same frequency in both time series are statistically significant and if their periodicities are related to each other [31]. This method can be used to identify the frequency response of complex systems, even in the presence of noise and artifacts.

The power correlation (POWCORR) is the correlation between two orthogonalized signals in frequency (co-variation frequency). The method starts by applying spectral analysis to the power envelopes of the signals, using a wavelet-based approach. The coherency between the power envelopes is then calculated, and the real part of this coherency is used as a measure of frequency-specific correlation [32].

**Parametric methods.** Compared to nonparametric methods, parametric methods are more widely accepted for estimating the effective connectivity of multi-channel EEGs. Granger causality (GC) [23] provides two estimates of directed connectivity between a given signal pair, measuring separately the influence of signal $x$ on signal $y$ and the influence of signal $y$ on signal $x$. GC can be applied in both the time and frequency domains. In this study, only the frequency domain is considered. The computation GC in the frequency domain requires the estimation of the spectral transfer matrix $H(\omega)$, and the cross-spectral density matrix $S(\omega)$ for signal pair $(x, y)$ at frequency $\omega$. It is obtained from:

$$GC_{xy}(\omega) = ln\left(\frac{S_{yy}(\omega)}{S_{yy}(\omega) - \left(\sum_{xx} - \frac{\sum_{yx}^2}{\sum_{yy}}\right)|H_{yx}(\omega)|^2}\right) \quad (3)$$

where $S(\omega) = H(\omega)\Sigma H(\omega)^*$.

The multivariate approach yields a spectral transfer matrix that can be used to compute a set of connectivity metrics, which are related to Granger causality: the directed transfer function (DTF) and partial directed coherence (PDC). These quantities are normalized between 0 and 1. The normalization factor is defined as the sum along the rows of the spectral transfer matrix for DTF and as the sum along the columns of the inverse of the spectral transfer matrix for PDC. PDC is computationally more efficient and more robust than DTF since it does not involve any matrix inversion [23].

## Microstate feature extraction

As functional connectivity matrices mainly extract spatial representation, the subsequent microstate analysis focuses on the representation in the time domain. This analysis employs clustering methods on the time domain of EEG signals to extract features within the temporal domain.

The procedure for microstate analysis follows the methodology detailed in our previous study [33]. The primary steps for microstate feature extraction include microstate segmentation and the calculation of statistical microstate parameters. The detailed steps are as follows:

**Microstate segmentation.** This section details the derivation of microstate prototypes from the combined dataset. Initially, the global field power (GFP), representing the spatial standard deviation of EEG signals across all channels, was calculated. Subsequently, a clustering method grouped the GFP sequences into a few classes based on topographic similarity. We employed a modified k-means method in this study, where each resulting cluster represented a

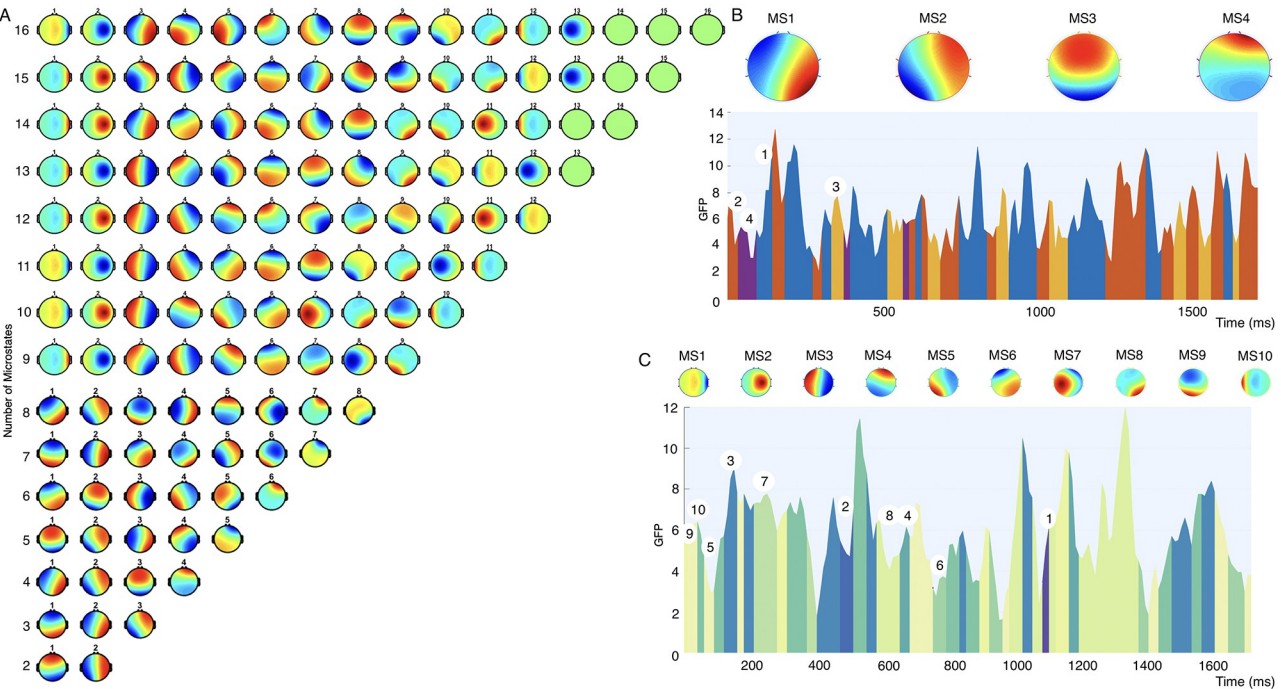

**Fig 3. The segmentation of EEG microstates and microstates prototypes.**

topographical prototype, known as a microstate prototype. The modified k-means method incorporates additional features into the clustering process [33, 34].

Theoretically, the number of clusters in the microstates segmentation can be any number. In previous studies, the number of four clusters is the most commonly explored method. Many studies focus on the analysis of the correlations between some phenotypes with the four common microstate prototypes (A, B, C and D). This study focuses on extracting statistical parameters from microstates and using them as features to input classifiers to classify different phenotypes. Therefore, the cluster number in this study is not limited to four.

We set the number of clusters from 2 to 30 in our experiments and Fig 3A shows the micro prototypes of cluster numbers from 2 to 16. Fig 3B and 3C reveals the segmentation of EEG signal based on GFP for four number of cluster and 10 number of cluster, respectively.

**Microstate parameters.**   Here, we extracted the following microstate parameters: The global field power (GFP), Duration, Occurrence, Coverage, and The global explained variance (GEV) [16, 35]. The detailed definitions of these parameters are provided in Table 2.

**Table 2. Microstate parameters.**

| Microstate parameters | The detailed definitions |
|---|---|
| GFP | The root mean square of the average-referenced electrode values at a given time instant. |
| Duration | The average time a given microstate remains stable. |
| Occurrence | The frequency with which a microstate occurs within one second. |
| Coverage | The total percentage of time a given microstate occupies. |
| GEV | How similar each EEG sample is to the microstate prototype it has been assigned to. |

## An ensemble model: RSBagging

To detect post-stroke conditions, extracting meaningful information from EEG signals and identifying effective biomarkers for various conditions are promising approaches. These can then be used with machine learning for patient classification. However, post-stroke conditions often represent a minority class, making them difficult to identify with standard classifiers like XGBoost and Random Forest. To overcome this challenge, we developed an ensemble classifier using a combination of bagging and random undersampling, referred to as RSBagging.

### Framework of RSBagging

Fig 4 illustrates the procedure and framework of our RSBagging method, which operates as follows: First, the original dataset is divided into a training set and a test set. The training set is then randomly undersampled into $N$ subsets. Each of these $N$ subsets is used to train an individual SVM classifier. Each SVM model makes its own classification decisions on the same test dataset. Finally, the classification results from the $N$ SVM models are combined using majority voting to produce the final classification decision.

In comparison to recent ensemble models like XGBoost and Random Forest, our RSBagging model introduces diversity through the random resampling of the training dataset. Diversity is essential for creating accurate ensembles, and each basic model in RSBagging is trained on a different subset of the full training set.

The RSBagging classifier addresses the classification of imbalanced datasets in two main steps: data splitting and resampling, model training and fusion. These steps are detailed below:

### Data splitting and resampling

To train and evaluate the RSBagging classifier, the original dataset $D_{original}$ is divided into a test set $D_{test}$ and a training set $D_{train}$ using stratified sampling. This method maintains the class

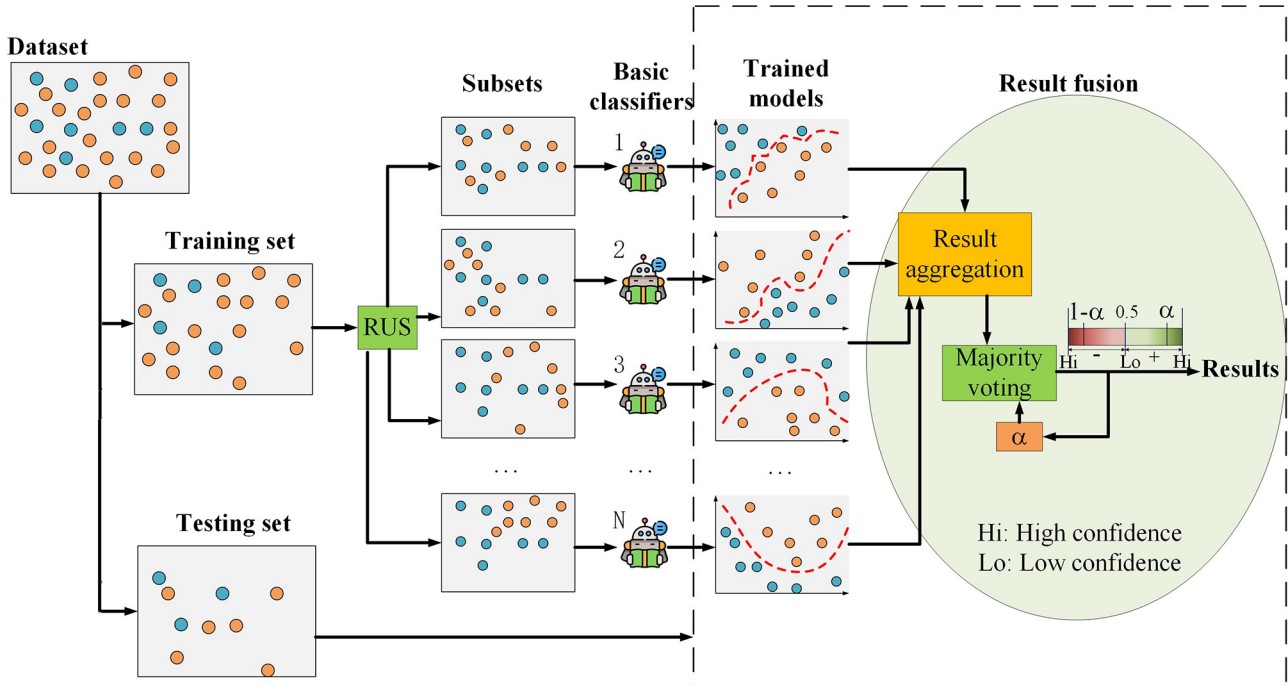

**Fig 4. The framework of our ensemble classifier: RSBagging model.**

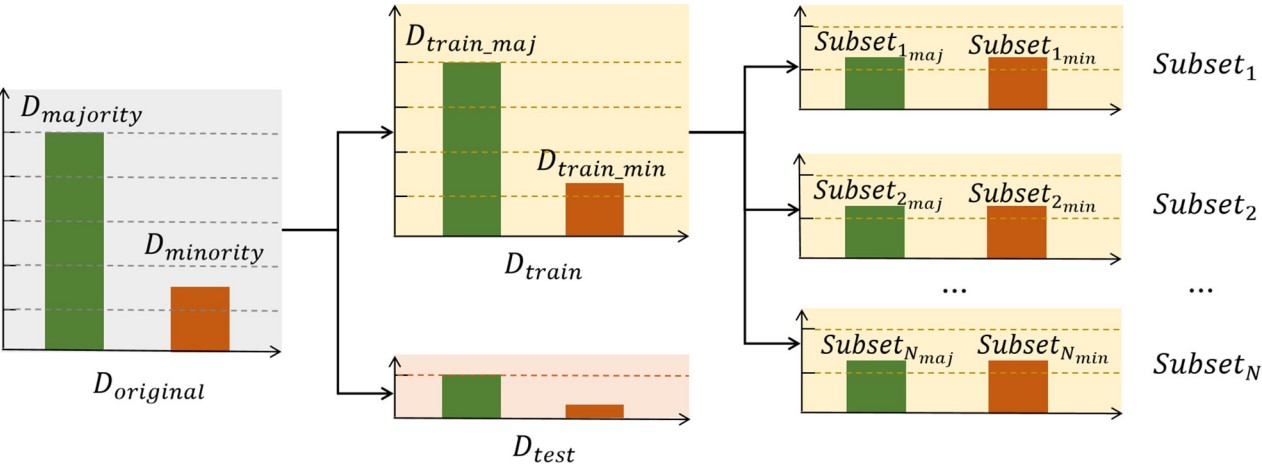

**Fig 5. The procedure of data spliting.**

ratio in the test set to match that of the original dataset, allowing the classifier to be evaluated on an imbalanced dataset similar to what is encountered in real-world clinical situations.

In the training phase, the training set is further divided into $N$ subsets for the $N$ basic models in the RSBagging classifier. Random undersampling is used to add diversity to the RSBagging ensemble classifier by creating different training subsets for each model. The test set is then input to all trained basic models in the test phase.

The process of splitting the training data can be formalized as follows: Given an imbalanced training dataset $D_{train}$ with $m$ samples ($|D_{train}| = m$), where each sample is an instance-label pair $(x_i, y_i)$ with $x_i$ in the n-dimensional feature space $X = f_1, f_2, \ldots, f_n$ and $y_i \in Y = 1, 2$ as its class label. The sets $D_{train_{min}}$ and $D_{train_{maj}}$ represent the minority and majority class samples in $D_{train}$, respectively, and $D_{train_{min}} \cup D_{train_{maj}} = D_{train}$ and $D_{train_{min}} \cap D_{train_{maj}} = \Phi$.

For each of the $N$ basic classifiers, a subset $Subset_n$ is generated and labeled, where $n = 1, \cdots, N$. The subsets $Subset_{n_{min}}$ and $Subset_{n_{maj}}$ in each $Subset_n$ represent the minority and majority samples, respectively.

To generate $Subset_n$, $D_{train_{min}}$ is used as $Subset_{n_{min}}$ in each of the $N$ subsets. Then, we construct its majority class sample set $Subset_{n_{maj}}$ of the same size as $D_{train_{min}}$ by randomly sampling $D_{train_{maj}}$, i.e., $|Subset_{n_{maj}}| = |D_{train_{min}}|$ and $Subset_{n_{maj}} \subset D_{train_{maj}}$. Finally, $Subset_n$ is created by combining $Subset_{n_{maj}}$ and $Subset_{n_{min}}$, resulting in $Subset_n = Subset_{n_{maj}} \cup Subset_{n_{min}}$, $n = 1, \cdots, N$. Each $Subset_n$ is input to a basic classifier in the RSBagging ensemble.

The data splitting process is illustrated in Fig 5.

## Model training and fusion

In our RSBagging ensemble classifier, we selected SVM as the base model. Each SVM model is trained on a distinct training subset, using a Gaussian kernel function. During training, each SVM optimizes the classification function by maximizing the margin between the training samples and the class boundary. The hyperparameters are tuned by maximizing this margin.

The final step in the RSBagging model involves merging the outputs from the individual base models to generate the overall result. This fusion process can be executed at various stages, such as during input feature processing, the training phase, or at the results level. In our

RSBagging model, fusion is performed at the final stage. We employ majority voting to combine the results from all SVM models. Given that hard voting does not yield optimal classification performance for our study, we use soft voting with a threshold $\alpha$. This threshold is an adjustable parameter in RSBagging, tailored to specific classification tasks and datasets, with values ranging from 0.5 to 1.

## Parameters

There are two parameters that shall be adjusted in the RSBagging model according to the distribution of a dataset. The first one is the number of basic SVM models in the RSBagging, $N$. Normally, the more heavily imbalanced the dataset is, the bigger the parameter $N$ shall be used in the EOSVM classifier. The second parameter in the RSBagging model is the threshold value in the result fusion, $\alpha$, which also reveals the level of confidence in the result fusion. Our experiments show different classification results when $\alpha$ is set with different levels.

## Evaluation metrics

We utilize five metrics to assess the classification performance of RSBagging: accuracy, sensitivity, specificity, F1-score, and Predict Rate. The first four metrics are standard measures in machine learning for evaluating classification results. In our evaluation, we designate the minority class in the dataset as positive cases and the majority class as negative cases.

Let $TP$, $TN$, $FP$, and $FN$ represent true positives, true negatives, false positives, and false negatives, respectively. The metrics of accuracy, sensitivity, specificity, and F1-score are mathematically defined as follows:

$$Accuracy = \frac{TP + TN}{TP + TN + FP + FN} \times 100\% \tag{4}$$

$$Sensitivity = \frac{TP}{TP + FN} \times 100\% \tag{5}$$

$$Specificity = \frac{TN}{TN + FP} \times 100\% \tag{6}$$

$$F1-Score = \frac{2TP}{2TP + FP + FN} \times 100\% \tag{7}$$

The final metric employed in this study is Predict Rate, which quantifies the percentage of subjects receiving either a True or False prediction from the RSBagging classifier for a given parameter $\alpha$. A higher Predict Rate indicates a higher number of subjects that the classifier can classify. This metric is essential here because as the parameter $\alpha$ increases in RSBagging, some subjects may receive no final prediction.

$$\text{Predict Rate} = \frac{TP + FP + TN + FN}{N^{\oplus} + N^{\ominus}} \times 100\% \tag{8}$$

where $N^{\oplus}$ and $N^{\ominus}$ denote the total numbers of positive and negative cases, respectively.

## Experimental results and discussions

### Disorder of consciousness recognition

We conducted experiments on classifying a stroke subject into one of two classes: positive with DoC, or negative without DoC. The stroke patients involve 97 subjects with DoC and 144 subjects without DoC. Two kinds of features (microstate statistic parameters and connectivity matrices) are input to classifiers to detect DoC in stroke patients.

**Classification from EEG microstates features.** Firstly, we use six existing classifiers to detect DoC in stroke patients. Table 3 shows the mean of different microstate cluster (from 2 to 28) classification result. It is seen from Table 3 that the existing six classifiers Adaboost (Ada), Random Forest (RF), K-nearest neighbors (KNN), Xgboosting (XGB), Support vector machine (SVM) and Decision tree (Tree) cannot effectively detect DoC in stroke patients. Despite some classifiers achieving an accuracy higher than 80%, the sensitivity is really poor (ranging from 0 to 52%) even when employing resampling methods like Random under-sampling (Rus) and Random over-sampling (Ros). The last two rows of Table 3 present the best performance of our RSBagging model, using microstate features (with a cluster number of 16) and the PPC connectivity measure, respectively.

Then, we use our RSBagging model to detect DoC in stroke patients. Fig 6 reveals the results of the experiment. The subfigures in the first row are the results (Accuracy in subfigure A, Sensitivity in subfigure B) from the microstate feature when EEG signals are clustered to the different number of microstates (from 2 to 28 shown in the $x$ axis). The different colour lines are the results of the different parameter $\alpha$ in our RSBagging. As we find the classification performance gets the best when the number of microstates clustered is 16. Therefore, we explore the classification performance when the cluster number is 16 and the accuracy, sensitivity, Predict Rate and F1-Score are shown in subfigures C, D, E and F, respectively. In the last

**Table 3. The classification results on the DoC recognition based on microstate features from the existing classifiers.**

| Resample | Model | Accuracy (%) | Specificity (%) | Sensitivity (%) | F1-Score |
|---|---|---|---|---|---|
| Original | Ada | 59.57 | 100.00 | 0.00 | 0.00 |
| | RF | 59.57 | 100.00 | 0.00 | 0.00 |
| | KNN | 59.57 | 100.00 | 0.00 | 0.00 |
| | XGB | 59.57 | 100.00 | 0.00 | 0.00 |
| | SVM | 68.23 | 88.72 | 38.03 | 0.48 |
| | Tree | 59.57 | 100.00 | 0.00 | 0.00 |
| Rus | Ada | 59.57 | 100.00 | 0.00 | 0.00 |
| | RF | 59.57 | 100.00 | 0.00 | 0.00 |
| | KNN | 59.57 | 100.00 | 0.00 | 0.00 |
| | XGB | 59.57 | 100.00 | 0.00 | 0.00 |
| | SVM | 69.01 | 87.24 | 42.14 | 0.51 |
| | Tree | 59.57 | 100.00 | 0.00 | 0.00 |
| Ros | Ada | 76.17 | 96.49 | 46.22 | 0.60 |
| | RF | 74.66 | 89.76 | 52.40 | 0.62 |
| | KNN | 68.52 | 79.62 | 52.16 | 0.57 |
| | XGB | 70.64 | 83.81 | 51.23 | 0.57 |
| | SVM | 75.59 | 96.14 | 45.32 | 0.59 |
| | Tree | 70.39 | 84.67 | 49.34 | 0.57 |
| RSBagging (Microstate 16) | | 100 | 100 | 100 | 1 |
| RSBagging (PPC) | | 99.04 | 100 | 97.63 | 0.98 |

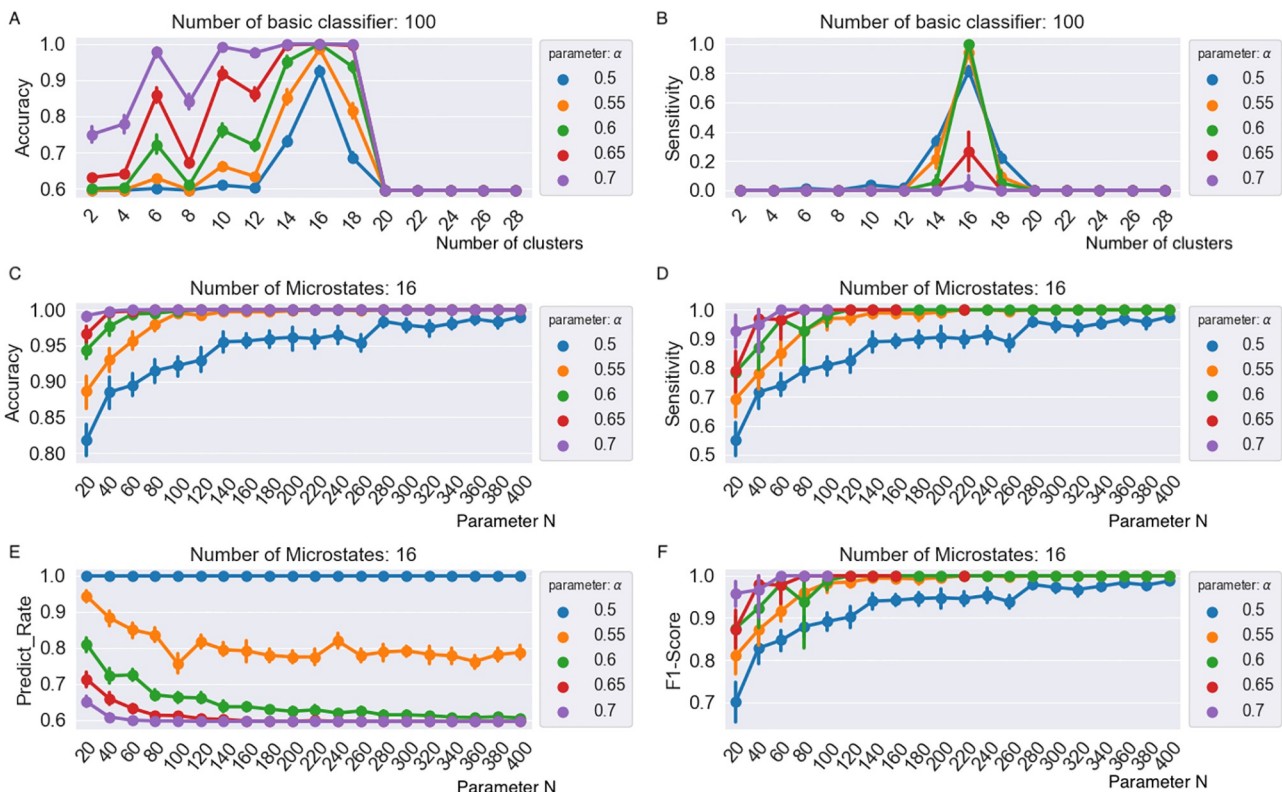

**Fig 6. The classification results on the DoC recognition based on microstate features from our RSBagging model.**

four subfigures, the *x* axis is the value of parameter *N* in our RSBagging. *N* represent the number of basic SVMs in our RSBagging. We can see that with the increase of *N*, the classification result from the number of microstates.

From Fig 6, it is seen that the number of microstate clusters and the parameter *N* of RSBagging all makes an impact on the classification result. From the comparison of different microstate features in the first two subfigures, we can see that the accuracy and sensitivity from clusters 14,16 and 18 are better than the others when parameter $\alpha$ in our RSBagging is set to 0.5. Further observation shows that with the increase of parameter $\alpha$, despite the accuracy increase, the sensitivity gets better only in the feature when the cluster number of the microstate is 16. From the last four subfigures, we can see that with the increase of parameter *N* in RSBagging, the accuracy, sensitivity, and F1-score rise (Fig 6C, 6D and 6F), however the Predict Rate decrease despite when $\alpha$ is 0.5 (Fig 6E).

**Classification from EEG connectivity matrices.** The three existing classifiers SVM, RF and XGB are used to detect DoC through 11 connectivity matrices. Fig 7 shows the results from the original dataset, the dataset after random undersampling and random oversampling. From the result from the original dataset (Fig 7A and 7B), it can be seen that the accuracy from the three classifiers reaches almost 70% while the sensitivity is poor (all below 60%). After resampling the imbalanced original dataset (Fig 7C and 7D), the sensitivity from the three existing classifiers increases significantly. However, the sensitivity from after random under-sampling and random over-sampling are still below 70%.

Fig 8 reveals the classification result from our RSBagging model. Fig 8A shows the classification result from 11 connectivity matrices with the parameter $\alpha = 0.5$. We can see that the

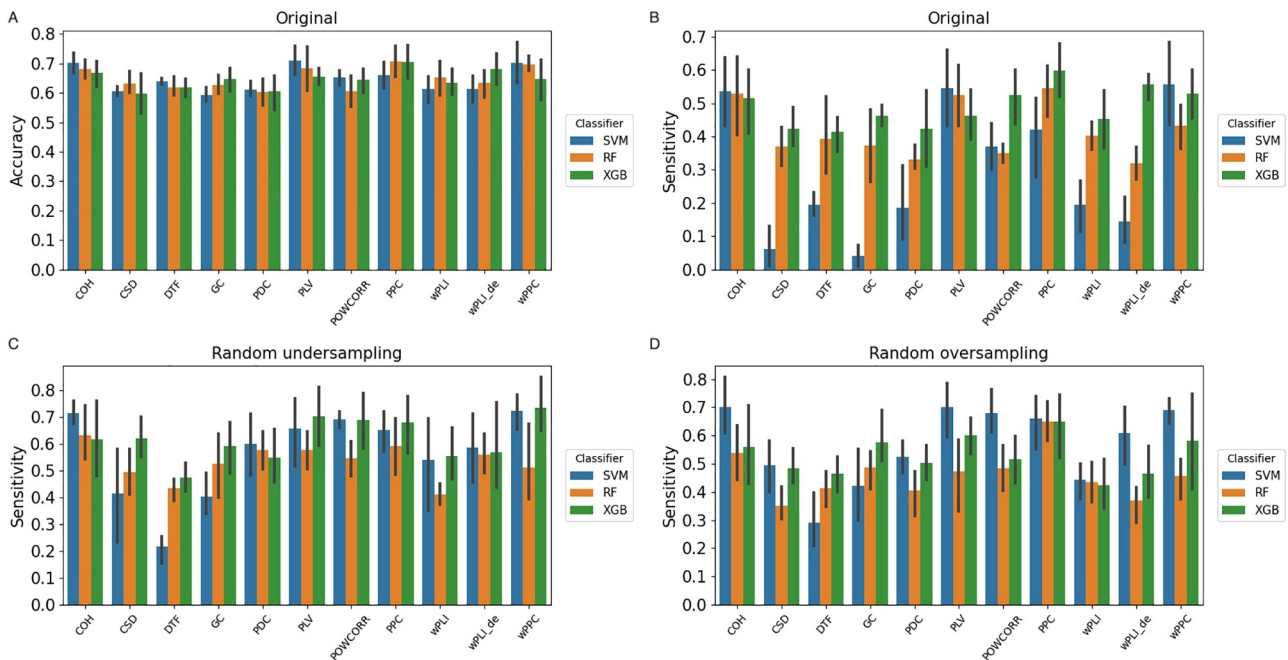

**Fig 7. The classification results on the DoC recognition based on 11 connectivity matrices from the existing classifiers.**

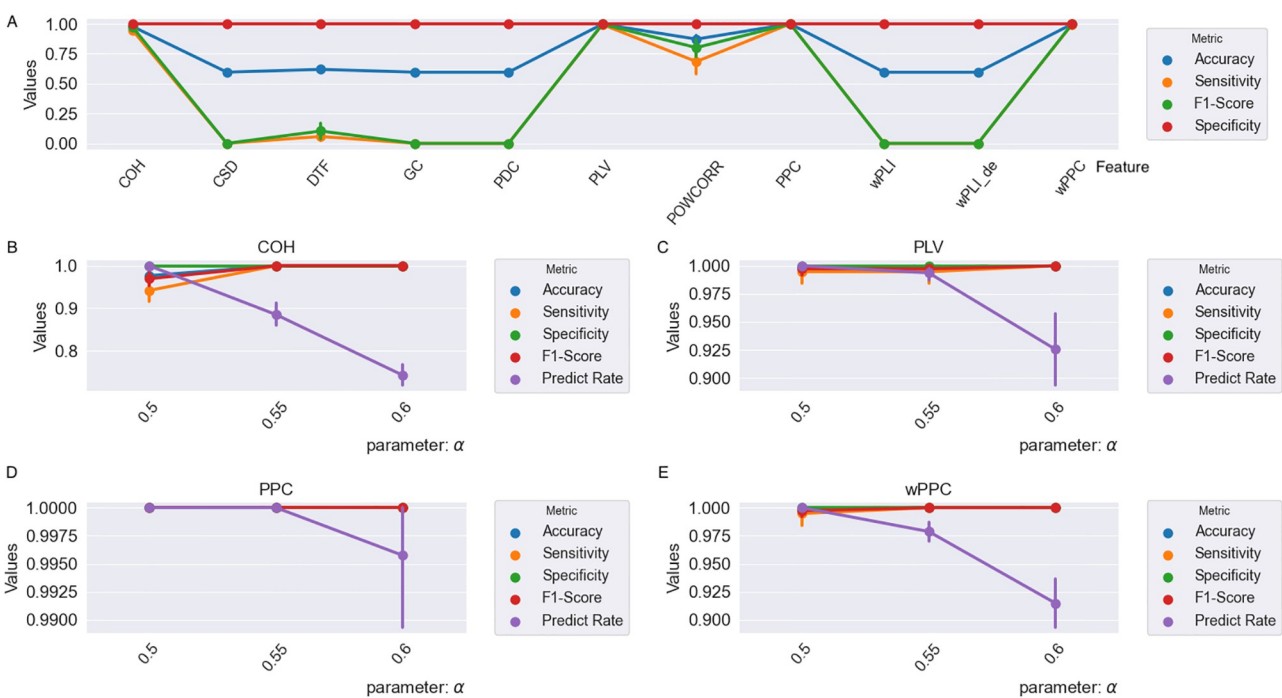

**Fig 8. The classification results on the DoC recognition based on 11 connectivity matrices from our RSBagging model.**

four connectivity matrices (COH, PLV, PPC and wPPC) get better performance than the others with the four metrics accuracy, sensitivity, specificity and F1-score all reaching higher than 99%.

Then, we adjust the parameter $\alpha$ in our RSBagging model from 0.5 to 0.6 to further explore the performance of the four connectivity matrices. Fig 8B shows the result from COH, Fig 8C shows the result from PLV, Fig 8D reveals the result from PPC and Fig 8E shows the result from wPPC. It can be seen that, with the increase of the parameter $\alpha$, despite the metrics Predict Rate decrease, the other four metrics increase. Among the four connectivity matrices, PPC gets the best performance with all the five metrics higher than 99.75% with the $\alpha$ being 0.5, PLV and wPPC are also effective features with all the five metrics higher than 98% with the $\alpha$ being 0.5.

### Motor disturbance classification

The second kind of classification task in this study is classifying stroke patients with different kinds of motor disturbances. When doing this task, we ruled out subjects with DoC as the state of consciousness impact on the EEG signal. Therefore, the stroke subject number in this section is 106 stroke patients subjects including 23 subjects with bilateral motor disturbance, 37 subjects with left motor disturbance, and 46 subjects with right motor disturbance.

This section completes two classification tasks: subjects with left motor disturbance (37) vs subjects with right motor disturbance (46), subjects with single motor disturbance (83) vs subjects with bilateral motor disturbance (23). The feature used in these two tasks is 11 connectivity matrices as our experimental results show that the microstate feature is not effective here.

**Classification from three existing classifiers.**   We use the three existing classifiers to complete the two classification tasks first. Fig 9 shows the experiment results. The first four

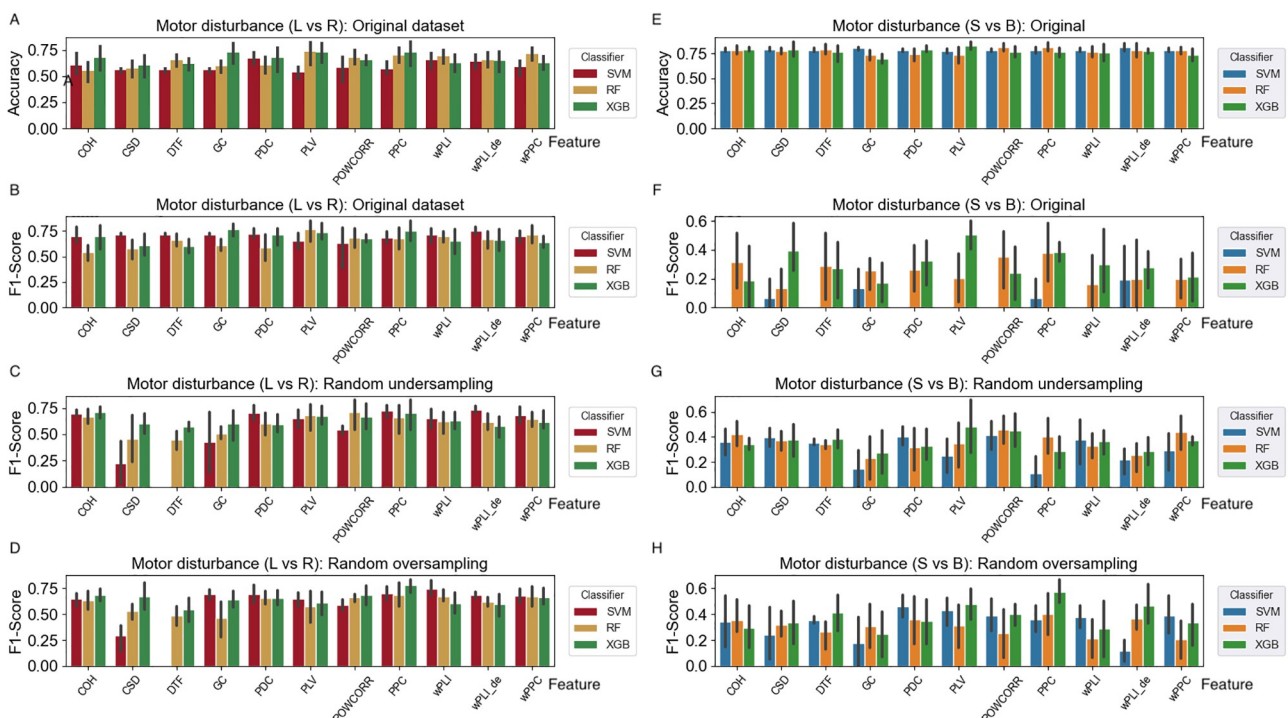

**Fig 9. The results on the classification of motor disturbance based on 11 connectivity matrices from the existing classifiers.**

subfigures Fig 9A–9D show the classification of subjects with left vs right motor disturbance and the last four subfigures Fig 9E–9H show the classification of subjects with single vs bilateral motor disturbance. Fig 9A and 9B reveal the classification accuracy and F1-score from the original dataset and Fig 9C and 9D show the F1-score of classification from dataset after random under sampling and over sampling, respectively. Fig 9E–9H is similar to the first four subfigures despite they are the result of the task of classification of subjects with single and bilateral motor disturbance.

From Fig 9, we can see that the accuracy and F1-score for the two tasks are all below 75% even after resampling the dataset to address the imbalanced problem. Especially, for the classification of subjects with single and bilateral motor disturbance, the F1-score is below 50% even after resampling the dataset.

**Classification from our RSBagging model.** As the two datasets in the classification of motor disturbance are also imbalanced, we try our RSBagging model on these two tasks. Fig 10 shows the experiment results. Fig 10A shows the classification results from 11 connectivity matrices for the classification of subjects with left vs right motor disturbance and Fig 10B shows the classification of subjects with single vs bilateral motor disturbance from 11 connectivity matrices. We can see that the three features COH, PLV and PPC get better performance than the other features in Fig 9A. In Fig 9B, the better features are GC, PDC and wPLI_de.

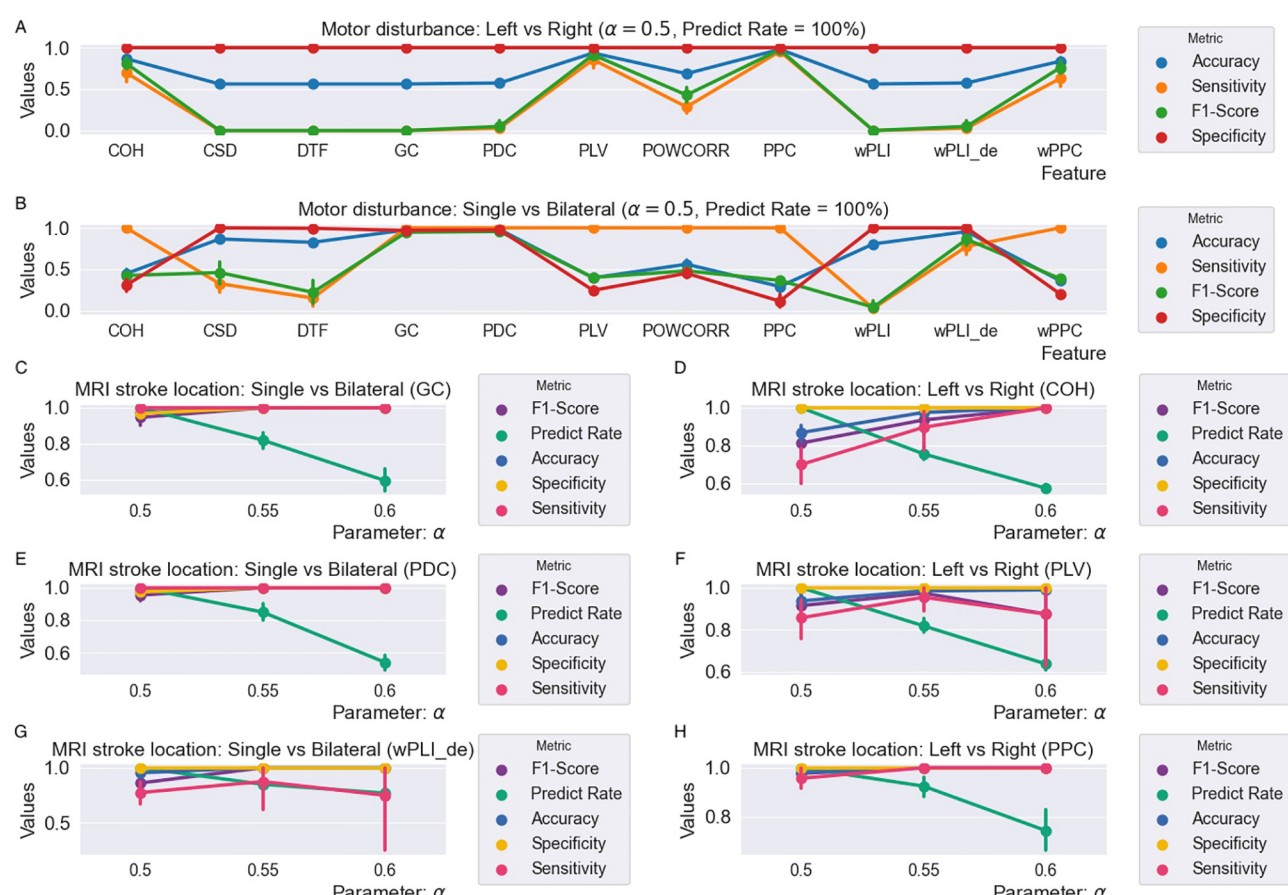

**Fig 10. The results on the classification of motor disturbance based on 11 connectivity matrices from our RSBagging model.**

To further explore the six connectivity matrices that perform better than the others in the above two tasks, we adjust the parameter $\alpha$ in our RSBagging model from 0.5 to 0.6. Fig 9C, 9E and 9G result from feature GC, PDC and wPLI_de, respectively for the classification of subjects with single and bilateral motor disturbance. Fig 9D, 9F and 9H are the result of the classification of subjects with left vs right from the feature COH, PLV and PPC, respectively.

From the last six subfigures in Fig 10, it can be seen that, with the rise of parameter $\alpha$, despite the reduction of metric Predict Rate, the other four metrics nearly all increase. More specifically, the features GC, PDC, COH, and PPC all get their best accuracy, sensitivity, specificity, and F1-Score above 99% although the Predict Rate reduces to around 60% or 80%. Therefore, we can adjust the parameter $\alpha$ according to the specific requirement of classification accuracy.

## Stroke location classification

The third kind of classification task in this study is classifying the subject with different stroke locations according to the subjects' MRI images. Similar to the task of motor disturbance, we ruled out subjects with DoC as the state of consciousness impact on the EEG signal. The dataset in this section contains 135 stroke subjects including 36 subjects with bilateral side stroke in the brain, 52 subjects with stroke in their left brain and 47 subjects with stroke in the right brain.

The features we used here are 11 connectivity matrices as our experimental results show that the Microstates features are not effective here. We complete two classification tasks: subjects with left brain (52) versus subjects with right brain (47), subjects with single brain side (99) versus subjects with bilateral brain side (36).

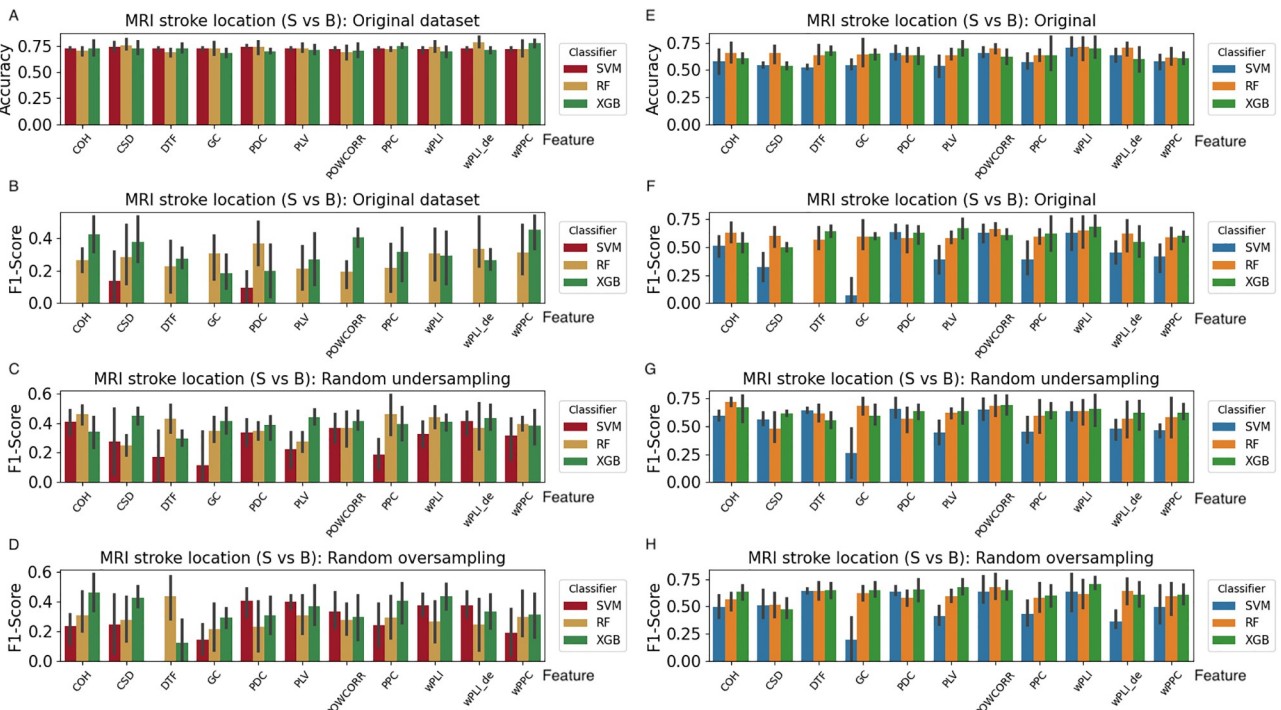

**Fig 11. The results on the classification of stroke location based on 11 connectivity matrices from the existing classifiers.**

Similar to the task of motor disturbance, we also employ the existing three classifiers to complete the two tasks about stroke location first. The experimental results are shown in Fig 11. It can be seen that the accuracy and F1-score are all below 75% even though the re-sampling method is used to address the imbalanced dataset.

Then, our RSBagging model is used to complete the two tasks referring to stroke location. The experimental results are shown in Fig 12. Fig 12A reveals the classification result from 11 connectivity matrices using RSBagging with $\alpha = 0.5$ for the task: stroke with left brain (52) vs stroke with the right brain. Fig 12B are results for the task of classifying subjects with the single brain side and subjects with the bilateral brain. Fig 12C and 12D are the results from a different parameter of $\alpha$ for two of the best connectivity matrices in Fig 12B.

From Fig 12, it is observed that the four connectivity matrices COH, PLV, PPC, and wPPC are effective in the task of classification of subjects with left-brain stroke and right-brain stroke with accuracy, sensitivity, specificity and F1-score all higher than 98%. For the task of classification of subjects with single brain side stroke and bilateral brain side stroke, the feature POWCORR is the most effective one reaching higher than 95% of accuracy, sensitivity, specificity and F1-score when the parameter of $\alpha$ is 0.5. The feature is the second effective feature for the same task with around 90% accuracy and higher 85% sensitivity, specificity and F1-score with $\alpha = 0.5$. From Fig 12D we can see when the $\alpha$ increase to 0.6, the accuracy,

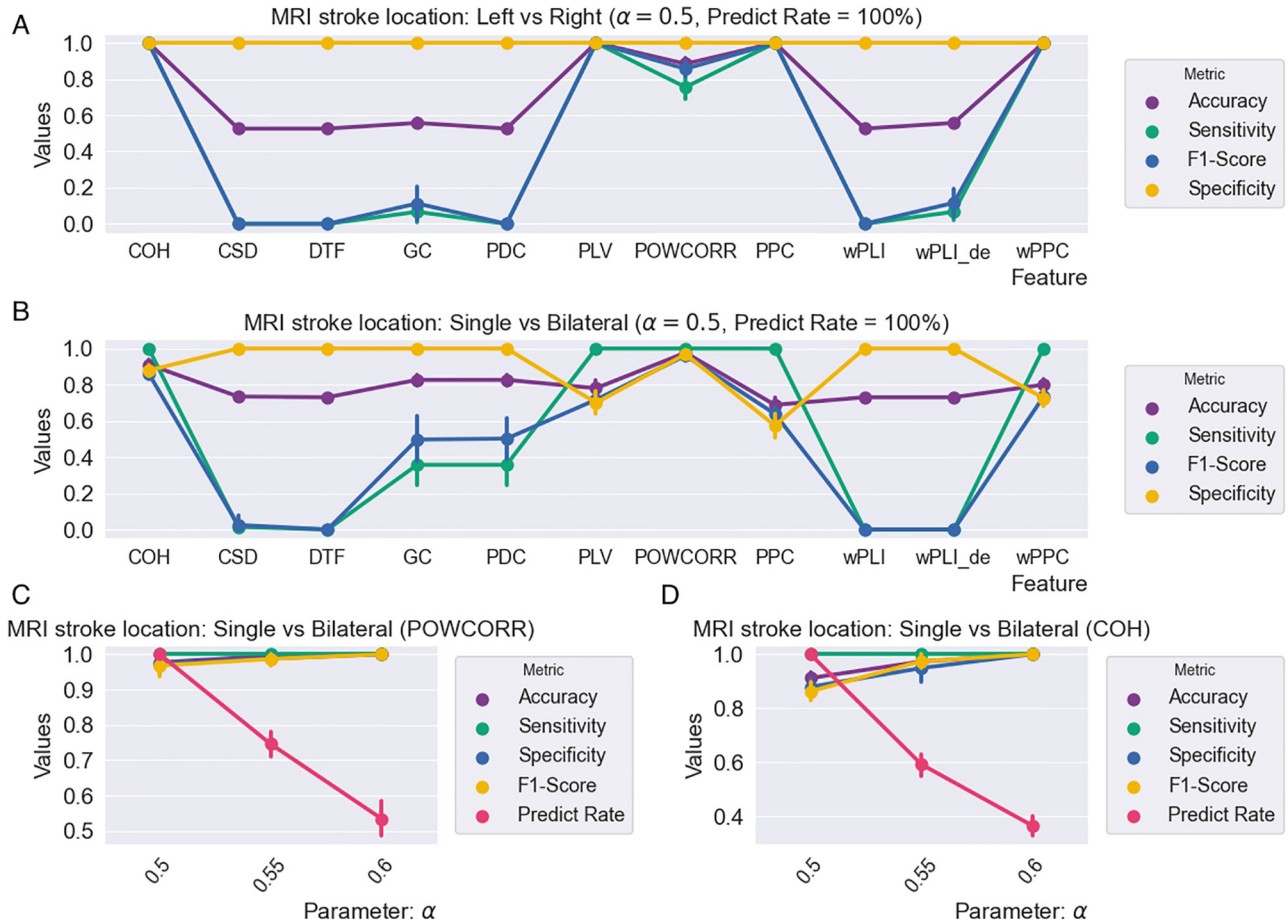

**Fig 12. The results on the classification of stroke location based on 11 connectivity matrices from our RSBagging model.**

sensitivity, specificity and F1-score all rise to higher than 98% however, the Predict Rate decrease to around 40%.

## Limitations

Despite the promising results of our EEG-based stroke classification, several limitations should be acknowledged. First, the dataset lacks detailed information on patients with multiple strokes, and the absence of longitudinal data prevents us from accounting for the effects of stroke recovery on brain activity, which may introduce confounding variables. Additionally, the lack of information on neurological recovery between strokes limits our ability to analyze how recovery trajectories affect EEG signals. Future studies should incorporate longitudinal data and more comprehensive medical histories, including stroke timing and frequency, to address these gaps. Finally, while our findings are insightful, the generalizability of the results would benefit from a larger, more diverse dataset. Future research should aim to validate these findings across different stroke subtypes and recovery stages with a broader population.

## Conclusion

In this study, we introduced the RSBagging model, an ensemble classifier, for the classification of the after-effects of stoke with imbalanced dataset. With this model, the classification results show that EEG connectivity matrices PPC, PLV, and wPPC are the three most effective biomarkers to monitor DoC in ischemic stroke patients. The connectivity matrices GC, PDC, and PPC are effective biomarkers that can reveal motor disturbance after stroke. For the classification of the stroke location in the brain, the connectivity matrices COH, PLV, PPC, wPPC and POWCORR are effective biomarkers. The statistical parameters from EEG microstates only show their effectiveness for the detection of DoC in ischemic stroke patients. Therefore, this study reveals the that EEG biomarkers can be used to monitor after-effects of ischemic stroke patients. In conclusion, this study reveals the that EEG biomarkers can be used to monitor after-effects with high classification performance on imbalanced datasets for ischemic stroke patients.

## Author Contributions

**Data curation:** Fang Wang, Peng Zhang, Fengyun Hu.

**Formal analysis:** Fang Wang.

**Investigation:** Fang Wang.

**Methodology:** Fang Wang.

**Project administration:** Fengyun Hu.

**Supervision:** Xueying Zhang.

**Writing – original draft:** Fang Wang.

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
