## [Decision Letter · Decision Letter 0]

9 Sep 2024

PONE-D-24-22084RSBagging: An Ensemble Classifier Detecting the after-effects of Ischemic Stroke through  EEG Connectivity and MicrostatesPLOS ONE

Dear Dr. Wang,

Thank you for submitting your manuscript to PLOS ONE. After careful consideration, we feel that it has merit but does not fully meet PLOS ONE’s publication criteria as it currently stands. Therefore, we invite you to submit a revised version of the manuscript that addresses the points raised during the review process.

We look forward to receiving your revised manuscript.

Kind regards,

Ramu Anandakrishnan, Ph.D.

Academic Editor

PLOS ONE

Additional Editor Comments:

Dear Dr. Wang,

This study presents an interesting approach for developing a classifier for predicting the after-effect of stroke. However, the reviewers have several questions, comments and concerns that need to be addressed. In addition, to these comments, please address the following:

1. Add a section describing the limitations of this study.

2. Compare the results for the various models using the training dataset versus the test dataset, discuss any differences and the implication for how the models might perform on an independent test set.

Thank you.

Reviewers' comments:

Reviewer's Responses to Questions

**Comments to the Author**

1. Is the manuscript technically sound, and do the data support the conclusions?

Reviewer #1: Yes

Reviewer #2: Yes

2. Has the statistical analysis been performed appropriately and rigorously? 

Reviewer #1: Yes

Reviewer #2: Yes

3. Have the authors made all data underlying the findings in their manuscript fully available?

Reviewer #1: Yes

Reviewer #2: Yes

4. Is the manuscript presented in an intelligible fashion and written in standard English?

Reviewer #1: Yes

Reviewer #2: Yes

5. Review Comments to the Author

Reviewer #1: The paper is well-written and the ensemble method combining SVM classifiers trained on balanced training subsets through undersampling produced impressive classification performance.

Some comments to improve readability of the paper:

1. Please state the shape of the data (how many total features per input, since there are pairwise connectivity metrics for each pair, is it quadratic on the electrode locations?)

2. It appears, building individual classifiers on balanced subsets through undersampling is the main reason behind the classifier performance. Please provide a discussion what would happen if the similar undersampling method is used to modify the random forrest or xgboost like methods.

3. Please make the figure 2 more comprehensible. For example in step 3, a pair of siganls was introduced, but in step 4, these terms were not used.

4. Please elaborate on the classification scheme and provide examples of topographic similarity for microstate clustering.

Reviewer #2: There should be some discussion/analysis on time from stoke to EEG as a confounding variable, as well as if there were multiple strokes separated by time as there might have been some degree of recovery after the initial stroke that could influence the results.

When comparing the different models, it would be helpful to have graph or table directly comparing the results of existing models vs the new RSbagging model, the current presentation obfuscates away a direct comparison and makes it difficult.

There are some small errors, SGD is not defined, other acronyms like COH, PLV are defined multiple times.

The line "The RSBagging classifier addresses the classification of imbalanced datasets in two mainly steps:..." should say main steps

Table 1 has inconsistent space formatting for the numbers in the rows, particularly with middle coma and deep coma have a space after the plus and minus and lack of a space after a slash

In results and discussion this part does not match table 1, which lists 23 subjects with bilateral motor disturbance: "Therefore, the stroke subject number in this section is 105 stroke patients subjects including 22 subjects with bilateral motor disturbance, 37 subjects with left motor disturbance, and 46 subjects with right motor disturbance."

This is also incorrect, and should say 47 subjects had a stroke in the right brain: "The dataset in this section contains 135 stroke subjects including 36 subjects with bilateral side stroke in the brain, 52 subjects with stroke in their left brain and 47 subjects with stroke in the left brain."

This is also incorrect, and should say 36 subjects with bilateral brain side: "We complete two classification tasks: subjects with left brain (52) versus subjects with right brain (47), subjects with single brain side (99) versus subjects with bilateral brain side (52)."

This should have a number for consistency: "...stroke with left brain (52) vs stroke with the right brain."

There should be a space here "...higher than98% however..."

This link to the data is incorrect: https://github.com/linda-edward/RSBagging/data.

6. PLOS authors have the option to publish the peer review history of their article (what does this mean?). If published, this will include your full peer review and any attached files.

Reviewer #1: No

Reviewer #2: No

---

## [Author Response · Author response to Decision Letter 0]

19 Sep 2024

We sincerely appreciate the time and effort you have dedicated to reviewing our manuscript. Your insightful comments have been invaluable in enhancing the quality and readability of the paper. We have carefully considered all of your feedback and implemented substantial revisions accordingly. We provide detailed responses in the response letter.

---

## [Editor Report · Decision Letter 1]

23 Sep 2024

RSBagging: An Ensemble Classifier Detecting the after-effects of Ischemic Stroke through  EEG Connectivity and Microstates

PONE-D-24-22084R1

Dear Dr. Wang,

We’re pleased to inform you that your manuscript has been judged scientifically suitable for publication and will be formally accepted for publication once it meets all outstanding technical requirements.

Kind regards,

Ramu Anandakrishnan, Ph.D.

Academic Editor

PLOS ONE
---

## [Editor Report · Acceptance letter]

9 Oct 2024

PONE-D-24-22084R1 

PLOS ONE

Dear Dr. Wang, 

I'm pleased to inform you that your manuscript has been deemed suitable for publication in PLOS ONE. Congratulations! Your manuscript is now being handed over to our production team.

Kind regards, 

on behalf of

Dr. Ramu Anandakrishnan 

Academic Editor

PLOS ONE